# Chronobiological analysis of sex differences in electrocardiographic parameters in spontaneously breathing Wistar rats under tiletamine-zolazepam anaesthesia

Viktória Novotná, Soňa Grešová, Pavol Švorc*

Department of Medical Physiology, Faculty of Medicine, Pavol Jozef Šafárik University, Košice, Slovak Republic

These authors contributed equally to this work.
* pavol.svorc@upjs.sk

## Abstract

### Introduction

General anaesthesia is essential in surgical interventions because it reduces stress and restricts the animal's movement. However, it can interfere with the circadian clock and, consequently, with several physiological functions, including the cardiovascular system. A combination of tiletamine and zolazepam is rarely used in rat studies; therefore, its effect on cardiovascular function in the context of sex and the light/dark cycle remains unknown.

### Aim

This study analysed the effect of sex and the light/dark cycle on electrocardiographic parameters in Wistar rats anaesthetised with tiletamine-zolazepam.

### Methods

Experiments were performed on spontaneously breathing Wistar rats of both sexes following a 4-week adaptation to a light/dark (12h/12h) cycle. After intraperitoneal administration of Zoletil 50 (30 mg/kg; Virbac; France), electrocardiographic parameters were measured in lead II using LabChart 8 (ID Instruments).

### Results

Regarding the effect of the light/dark cycle on electrocardiographic parameters, males showed a higher heart rate and a shorter PR interval during the dark period, whereas females exhibited a longer QRS interval and a higher R wave amplitude in the dark period than in the light period. Sexual dimorphism was present during the light period, with males showing a longer QT interval and females displaying a higher

**Data availability statement:** All data generated or analysed during this study are included in this manuscript and in the Supporting Information file (S1 Table).

**Funding:** This research was funded by the EU NextGenerationEU through the Recovery and Resilience Plan for Slovakia under the project No. 09I03-03-V05-00008 (to V.N.); https://crz.gov.sk/contract/9717095/. The funders had no role in study design, data collection and analysis, decision to publish, or preparation of the manuscript.

**Competing interests:** The authors have declared that no competing interests exist.

**Abbreviations:** ANOVA, analysis of variance; ECG, electrocardiography; HR, heart rate; LD cycle, light/dark cycle.

T wave amplitude. In the dark period, sex differences were observed only in the PR interval, which was shorter in males.

## Conclusion

The obtained results indicate a significant effect of sex and light/dark cycle on cardiovascular parameters in rats under tiletamine-zolazepam.

## Introduction

Rats are considered as one of the most suitable animal models for studying the cardiovascular system. In the context of electrocardiography (ECG), although certain differences exist between rats and humans, fundamental similarities remain. These similarities have supported the widespread use of rat ECG in basic cardiovascular research, particularly for evaluating cardiac function under physiological conditions and in experimental models of cardiovascular disease [1].

Except in telemetry-based studies, experiments on rats are commonly conducted under general anaesthesia. It is therefore important to consider the effects of general anaesthesia when interpreting ECG parameters [2]. General anaesthesia enhances parasympathetic activity, while sympathetic and baroreflex activity are suppressed [3]. During surgical intervention, these effects should be minimised to maintain the subject's capacity to respond to physiological changes [4]. Experimental research thus requires the selection of an appropriate anaesthetic and a clear understanding of its influence on the cardiovascular system. The importance of this knowledge is evident because all ECG parameters serve as key indicators of cardiac rhythm disturbances.

Circadian rhythms play a key role in regulating cardiovascular responses to ensure optimal function in alignment with daily behavioural and environmental cycles, such as the light/dark (LD) cycle alternation. This system adapts to varying levels of activity during wakefulness as well as recovery during sleep. Diurnal variability in several cardiovascular parameters is now well established, including blood pressure [5–7], heart rate (HR) [5], circulating catecholamines [8], blood coagulation markers [9], and vascular endothelial function [10]. Several studies also suggest an increased incidence of cardiovascular events in the early morning hours, including stroke [11,12], myocardial infarction [13–15], ventricular arrhythmias [16], and sudden cardiac death [15,17], some of which have been associated with changes in ECG parameters such as the RR, PR, QRS, and QT intervals [18].

Disregarding the circadian variability of electrophysiological parameters, along with other factors such as animal handling, sex, and additional variables, can lead to misinterpretation in *in vivo* experimental studies. For example, male rats with heart failure with preserved ejection fraction exhibit poorer survival and a higher risk of sudden death than do females [19]. From a chronobiological perspective, significant diurnal variability in HR and blood pressure has been observed in normotensive Wistar-Kyoto and spontaneously hypertensive rats [20]. However, it

remains an open question whether sex-related differences in ECG parameters exist in rats depending on the period of the LD cycle.

A combination of tiletamine–zolazepam (Zoletil) is less frequently used in rat research but more commonly applied in domestic and wild animals. Nevertheless, due to its minimal influence on vital functions and ECG rhythmicity [21], tiletamine–zolazepam may represent a suitable anaesthetic choice for rats in future cardiovascular and chronobiological studies. Despite this, its detailed effects on ECG parameters in rats remain insufficiently characterised. This study was therefore performed to analyse sex differences in ECG parameters of Wistar rats under tiletamine-zolazepam with respect to the effect of the LD cycle.

The obtained results indicate a significant effect of both factors, suggesting that these findings may be useful for further research on the safety profile of this anaesthetic, optimisation of the timing of surgical procedures to minimise cardiovascular risks, and improved management of physiological responses under general anaesthesia.

## Materials and methods

### Ethics statement

The study protocol complied with the Guide for the Care and Use of Laboratory Animals, published by the United States National Institutes of Health (NIH publication no. 85−23, revised 1996), and was approved by the Ethics Committee of the Faculty of Medicine at Pavol Jozef Šafárik University in Košice, Slovak Republic (permit no. 1A/2024). Electrocardiographic measurements were performed in Wistar rats under general anaesthesia using Zoletil 50 (30 mg/kg; Virbac, France). All efforts were made to minimise animal suffering throughout the experimental procedures.

### Animals and light conditions

Experiments were performed on normotensive Wistar rats of both sexes (body weight, 340 ± 40 g; age, 3–4 months) following a 4-week adaptation to a 12-h/12-h LD cycle. Environmental conditions in the animal facility were maintained at a relative humidity of 40%–60%, a cage temperature of 24°C, and an artificial light intensity of 80 lux during the light period of the LD cycle. Two animals were housed per cage, with unrestricted access to a standard pelleted diet and water *ad libitum*. Animal handling was carried out by qualified personnel. The rats were randomly divided into four experimental groups (n = 20–21 per group) according to sex and period of the LD cycle, in which ECG parameters were measured after tiletamine-zolazepam administration (males–light, females–light, males–dark, females–dark). The effect of the light period on the studied parameters was assessed after adaptation to an LD cycle with the light period between 06:00 and 18:00 h. The impact of the dark period was evaluated following adaptation to a reversed LD cycle, with the light period occurring between 18:00 and 06:00 h. Experiments were conducted continuously throughout the year, and results were averaged irrespective of the season and, in females, independently of the oestrous cycle.

### Experimental design

In the adaptation room, prior to tiletamine-zolazepam administration, the animals were weighed to calculate the appropriate intraperitoneal dose of Zoletil 50 (30 mg/kg; a combination of tiletamine-zolazepam; Virbac, France), and their rectal temperature was measured. Individual animals were anaesthetised withtiletamine-zolazepam at 09:00 and 12:00 h (Fig 1). The depth of general anaesthesia was assessed by the loss of the righting reflex and the response to a painful stimulus, indicated by minimal limb movement or changes in muscle tone. Subsequently, the animal was transferred to the operating room, where it was positioned supine on the experimental table. Because the experiments were performed at the body temperature measured prior to tiletamine-zolazepam administration, rectal temperature was measured again after transfer and adjusted to the original value using an infrared lamp (a drop in rectal temperature of 1.5°C–2.0°C was typically observed approximately 20 minutes after tiletamine-zolazepam administration). For ECG recording, electrodes were

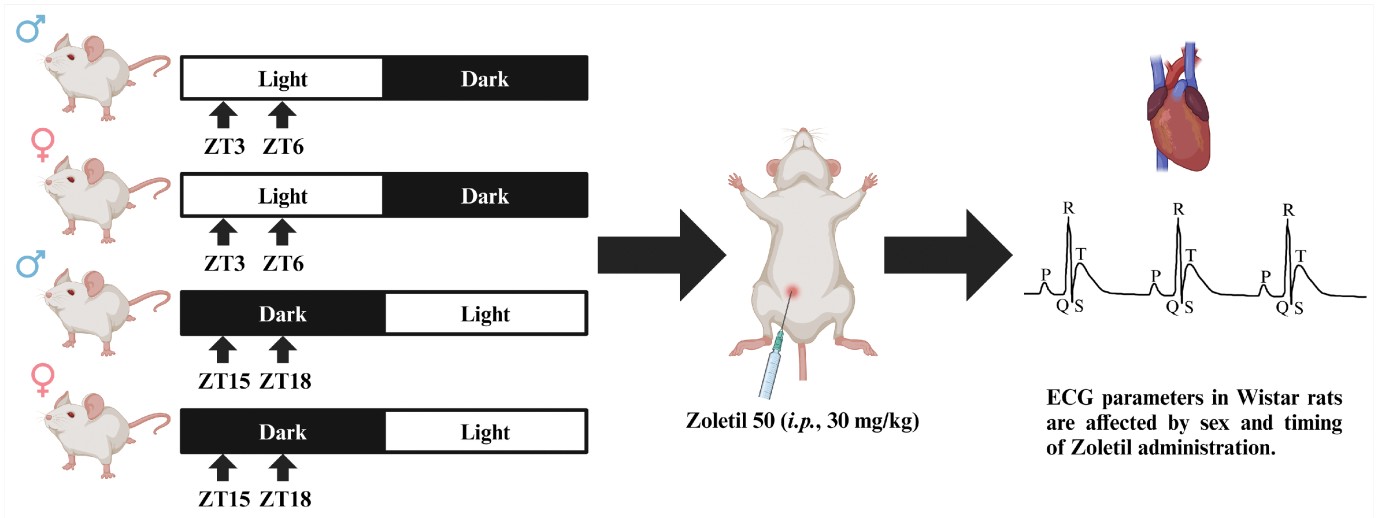

**Fig 1. Experimental design.** After a 4-week adaptation to a 12-h/12-h LD cycle, Wistar rats were randomly divided into four groups (n = 20-21 per group) according to sex and period of the LD cycle, in which ECG parameters were measured following Zoletil 50 administration (i.p., 30 mg/kg; a combination of tiletamine-zolazepam; Virbac; France). Arrows represent ZT; i.e., the time of tiletamine-zolazepam administration relative to the beginning of the light period (ZT0) in the animal facility. Because lights were switched on at 06:00 (ZT0) and off at 18:00 (ZT12), experiments performed at 09:00 and 12:00 correspond to ZT3 and ZT6 during the light period and to ZT15 and ZT18 during the dark period in a reversed LD cycle where the light period began at 18:00 (ZT0) and ended at 06:00 (ZT12). LD cycle – light/dark cycle; ZT – Zeitgeber time. Created with BioRender.com.

applied subcutaneously, and the depth of anaesthesia was reassessed using the same criteria—loss of motor activity (minimal limb movement, changes in muscle tone) or cardiovascular responses to painful stimuli (alterations in HR or the onset of cardiac arrhythmias). ECG parameters (HR, PR interval, QT interval, QTc interval with Bazett's correction, QRS interval, and the amplitudes of the P, R, and T waves) were recorded in the bipolar II limb lead in the dorsal position during spontaneous breathing and analysed with LabChart 8 software (ID Instruments). At the end of the experiments, animals were euthanised by intracardiac administration of a lethal dose of pentobarbital.

## Statistical methods

Two-way analysis of variance (ANOVA) was used to analyse the effect of two independent factors (sex and LD cycle periods) on ECG parameters, as well as to assess their interaction. Subsequently, ECG parameters were compared between individual groups (males–light, females–light, males–dark, females–dark) using Sidak's post hoc test. A *p*-value of <0.05 was considered statistically significant. Statistical analysis and graph generation were performed using Prism 8 software (GraphPad, San Diego, CA, USA).

In cases where a specific ECG parameter could not be clearly identified from the recording, it was marked as not measurable (NA) and excluded from the corresponding analysis (see S1 Table for raw ECG data). No additional outliers were excluded from the analysis. The number of animals per group (n = 20–21) was considered sufficient based on a power analysis conducted using the parameters: effect size f = 0.447 (estimated from empirical data), α error probability = 0.05, and power (1 – β) = 0.95.

## Results

The mean values of ECG parameters (HR, PR interval, QT interval, QTc interval with Bazett's correction, QRS interval, and the amplitudes of the P, R, and T waves) for each experimental group are presented in Table 1. The results of the

**Table 1. Mean values of ECG parameters in Wistar rats under tiletamine-zolazepam anaesthesia with 95% confidence intervals.**

| Parameter | Females–light | | Males–light | | Females–dark | | Males–dark | |
|---|---|---|---|---|---|---|---|---|
| | Mean±SD | 95% CI | Mean±SD | 95% CI | Mean±SD | 95% CI | Mean±SD | 95% CI |
| HR (beats/min) | 432.4 ±71.9 | 399.7–465.1 | 415.1±24.2 | 403.8–426.5 | 447.3±28.8 | 433.8–460.7 | 482.0±19.3 | 472.9–491.0 |
| PR interval (ms) | 50.5±8.5 | 46.0–55.0 | 54.9±3.3 | 53.4–56.4 | 53.3±9.5 | 48.7–57.8 | 42.6±3.1 | 41.1–44.1 |
| QT interval (ms) | 67.9±8.9 | 63.8–72.0 | 79.6±13.1 | 73.7–85.6 | 68.6±10.8 | 63.6–73.7 | 70.8±11.6 | 65.4–76.2 |
| QTc interval (ms) | 111.1±22.0 | 101.1–121.1 | 123.1±19.5 | 114.2–132.0 | 108.4±18.7 | 99.7–117.2 | 120.1±19.6 | 110.9–129.3 |
| QRS interval (ms) | 24.8±3.8 | 23.0–26.5 | 26.1±3.4 | 24.6–27.6 | 29.1±3.7 | 27.4–30.8 | 27.6±4.6 | 25.4–29.7 |
| P wave (µV) | 140.0±25.9 | 126.3–153.8 | 125.0±18.8 | 116.4–133.5 | 126.1±20.3 | 116.3–135.9 | 128.4±28.4 | 114.8–142.1 |
| R wave (µV) | 531.4±107.5 | 482.4–580.3 | 549.6±128.9 | 490.9–608.36 | 652.9±144.0 | 585.5–720.3 | 633.41±140.5 | 567.7–699.2 |
| T wave (µV) | 98.1±28.2 | 84.9–111.2 | 76.1±21.9 | 65.8–86.3 | 77.9±24.3 | 66.5–89.3 | 74.5±24.1 | 63.2–85.7 |

Data are expressed as mean±SD for a given experimental group (n=20–21 per group). CI – confidence interval; HR – heart rate; SD – standard deviation.

statistical analysis are summarised in Table 2, and the key findings of the study are outlined in Table 3. Representative ECG traces recorded under tiletamine–zolazepam from one animal in each experimental group (female–light, female–dark, male–light, male–dark) are shown in S2–S5 Figs. Due to the high interindividual variability in rats, even within the same experimental group, representative traces may not clearly display all significant effects.

## HR

HR in Wistar rats under tiletamine-zolazepam was significantly affected by the LD cycle and its interaction with sex. Males exhibited a higher HR during the dark period than during the light period, whereas in females, the difference between LD cycle periods was eliminated (Fig 2).

**Table 2. Results of statistical analysis.**

| Statistical test | Parameter | HR (beats/min) | PR interval (ms) | QT interval (ms) | QTc interval (ms) | QRS interval (ms) | P wave (µV) | R wave (µV) | T wave (µV) |
|---|---|---|---|---|---|---|---|---|---|
| Two-way ANOVA | *psex* | 0.3561 | **0.0457** | **0.0063** | **0.0090** | 0.9029 | 0.2461 | 0.9826 | **0.0241** |
| | *PLD cycle* | **<0.0001** | **0.0030** | 0.1063 | 0.5240 | **0.0011** | 0.3389 | **0.0006** | 0.0528 |
| | *pinteraction* | **0.0070** | **<0.0001** | 0.0575 | 0.9733 | 0.1017 | 0.1136 | 0.5157 | 0.0980 |
| | $F_{sex}$ (DFn, DFd) | (1.77) = 0.8620 | (1.77) = 4.137 | (1.78) = 7.883 | (1.78) = 7.174 | (1.78) = 0.01499 | (1.71) = 1.368 | (1.78) = 0.0004805 | (1.76) = 5.297 |
| | $F_{LD cycle}$ (DFn, DFd) | (1.77) = 18.93 | (1.77) = 9.479 | (1.78) = 2.671 | (1.78) = 0.4098 | (1.78) = 11.45 | (1.71) = 0.9272 | (1.78) = 12.64 | (1.76) = 3.869 |
| | $F_{interaction}$ (DFn, DFd) | (1.77) = 7.674 | (1.70) = 23.57 | (1.78) = 3.718 | (1.78) = 0.001127 | (1.78) = 2.743 | (1.71) = 2.566 | (1.78) = 0.4263 | (1.76) = 2.807 |
| Sidak's post hoc test (*p*-value) | F-L vs. F-D | 0.8413 | 0.7851 | >0.9999 | 0.9987 | **0.0038** | 0.4083 | **0.0231** | 0.0696 |
| | M-L vs. M-D | **<0.0001** | **<0.0001** | 0.0801 | 0.9976 | 0.7843 | 0.9979 | 0.2340 | >0.9999 |
| | F-L vs. M-L | 0.7258 | 0.2819 | **0.0066** | 0.2915 | 0.8555 | 0.2951 | 0.9982 | **0.0370** |
| | F-D vs. M-D | 0.0652 | **<0.0001** | 0.9906 | 0.3464 | <0.7709 | 0.9998 | 0.9978 | 0.9984 |

Results of the two-way ANOVA are presented as *p*-values for the effects of two independent factors and their interaction on ECG parameters in Wistar rats ($p_{sex}$ – effect of sex; $p_{LD\ cycle}$ – effect of LD cycle periods; $p_{interaction}$ – interaction of sex and LD cycle period). The table also includes F-test values comparing between-group and within-group variability ($F_{sex}$ – for sex; $F_{LD\ cycle}$ – for LD cycle periods; $F_{interaction}$ – for interaction between sex and LD cycle periods). Results of Sidak's post hoc test used to compare differences between groups (n=20–21 per group) are also expressed as *p*-values. Statistically significant results (*p*<0.05) are highlighted in bold. DFd – degrees of freedom denominator; DFn – degrees of freedom numerator; F-D – females–dark; F-L – females–light; HR – heart rate; M-D – males–dark; M-L – males–light.

Table 3. Key findings of the study.

| Category | Females | Males |
|---|---|---|
| Loss of LD differences | HR, PR interval | R wave amplitude |
| Maintenance of LD differences | R wave amplitude (↑ in dark) | HR (↑ in dark), PR interval (↑ in light) |
| Sex differences in light period | QT interval (↓), T wave amplitude (↑) | QT interval (↑), T wave amplitude (↓) |
| Sex differences in dark period | PR interval (↑) | PR interval (↓) |

The table presents the key conclusions of the study drawn from the statistical analysis. An upward arrow (↑) indicates an increased or prolonged value of the respective ECG parameter, while a downward arrow (↓) indicates a decreased or shortened value. HR – heart rate; LD – light/dark.

## Atrial complex

Statistical analysis revealed a significant effect of sex, LD cycle, and their interaction on PR interval duration. In males, the PR interval was shorter in the dark period than in the light period of the LD cycle. In females, the opposite but statistically nonsignificant tendency was detected. Moreover, in the dark period, males exhibited a shorter PR interval than females.

The amplitude of the P wave was not significantly affected by sex, LD cycle, or their interaction. No significant differences were observed between experimental groups (Fig 3).

## Ventricular complex

Analysis of the QRS interval duration revealed a significant effect of the LD cycle, with females specifically exhibiting a longer QRS interval during the dark period of the LD cycle.

The QT interval was significantly affected by sex, and the interaction between sex and LD cycle approached statistical significance. In the light period, the duration of the QT interval was significantly longer in males than in females.

Similarly to the QT interval, the QTc interval was also affected by sex (. However, the post hoc analysis did not reveal any statistically significant differences in the QTc interval duration between the groups.

The amplitude of the R wave was significantly influenced by the LD cycle, with females showing a higher R wave amplitude during the dark period than during the light period. In males, this trend was similar but not statistically significant.

The amplitude of the T wave was significantly affected by sex, while the effect of the LD cycle approached statistical significance. In the light period of the LD cycle, females exhibited a higher T wave amplitude than did males, which was not observed for the dark period (Fig 4).

## Discussion

The administration of general anaesthesia is essential to comply with the '3Rs' principles because it reduces stress, pain, and movement of the animal, but it may also disrupt the circadian rhythms of several physiological parameters, including HR and blood pressure, which can negatively affect the interpretation of results [22]. Our study suggests that females and males may respond differently to tiletamine-zolazepam in the context of maintaining circadian rhythms in HR. Previous studies have confirmed the tachycardic effect of tiletamine-zolazepam [23], whereas our observations indicate that the HR of males under tiletamine zolazepam is probably not dependent on the autonomic nervous system but is influenced by other factors [24].

P wave analysis has clinical relevance in humans; however, high-quality reference data on its morphological and temporal changes in rats are still lacking [1]. According to the literature, the absence of the P wave is associated with atrial fibrillation in both humans and rats [25,26], and its prolongation may increase the risk of supraventricular arrhythmias in Wistar rats, particularly following myocardial infarction [27]. Our study did not reveal any effect of sex or the LD

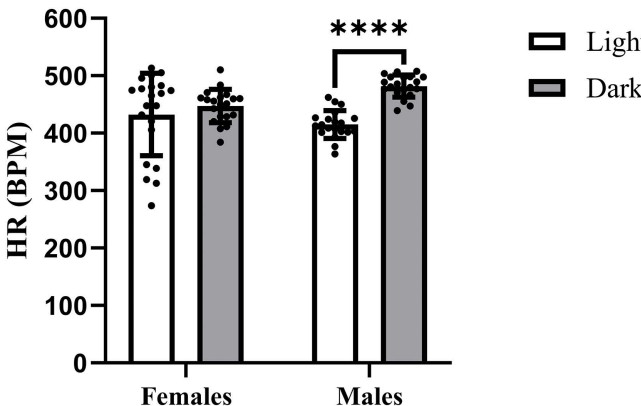

**Fig 2. Effect of tiletamine-zolazepam on HR in Wistar rats depending on sex and LD cycle.** Differences in HR are presented as individual data points, with columns representing the mean±SD for each group (white column – light period of the LD cycle; grey column – dark period of the LD cycle). Asterisks indicate the level of statistical significance according to Sidak's post hoc test: *$p < 0.05$; **$p < 0.01$; ***$p < 0.001$; ****$p < 0.0001$. HR – heart rate.

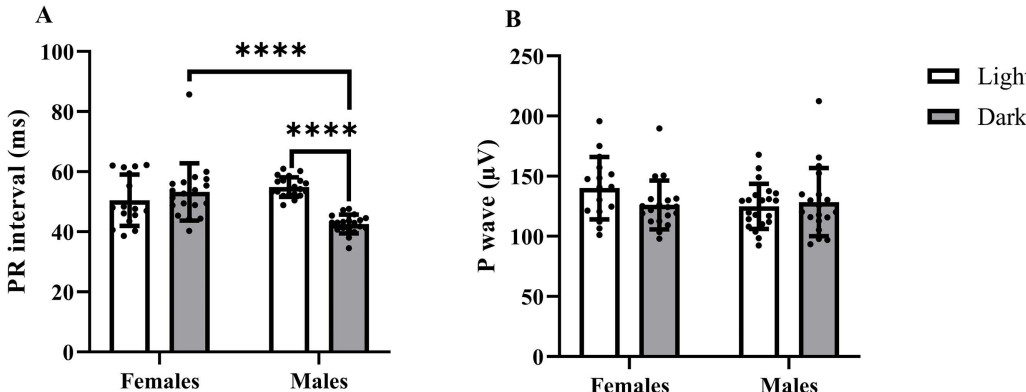

**Fig 3. Effect of tiletamine-zolazepam on the atrial complex in Wistar rats depending on sex and LD cycle.** Differences in **(A)** PR interval duration and **(B)** P wave amplitude are presented as individual data points, with columns representing the mean±SD for each group (white column – light period of the LD cycle; grey column – dark period of the LD cycle). Asterisks indicate the level of statistical significance according to Sidak's post hoc test: *$p < 0.05$; **$p < 0.01$; ***$p < 0.001$; ****$p < 0.0001$.

cycle on P wave amplitude, which is consistent with our previous findings in female Wistar rats [28]. However, we did not evaluate morphological changes of the P wave or other pathological alterations, as the aim of our study was the numerical description of ECG parameter changes in relation to sex and the LD cycle. Such an analysis could be beneficial in future investigations. By contrast, the duration of the PR interval was influenced by sex, the LD cycle, and their interaction. These changes suggest circadian variability in atrioventricular conduction in Wistar rats under tiletamine-zolazepam, depending on sex and the LD cycle, although further investigation is needed.The duration of the QRS complex reflects the time required for ventricular depolarisation and plays an important role in the evaluation of cardiac rhythm disturbances. In non-anaesthetised Wistar rats, the QRS interval duration ranges from 12 to 26 ms, with an average value of 17 ms [29]. Administration of tiletamine-zolazepam in Wistar rats prolonged the QRS interval, particularly during the dark period of the LD cycle in females, while in males it eliminated fluctuations in both QRS interval duration and R wave amplitude between the light and dark periods of the LD cycle. In females, however, rhythmicity in R wave amplitude was preserved. The QT

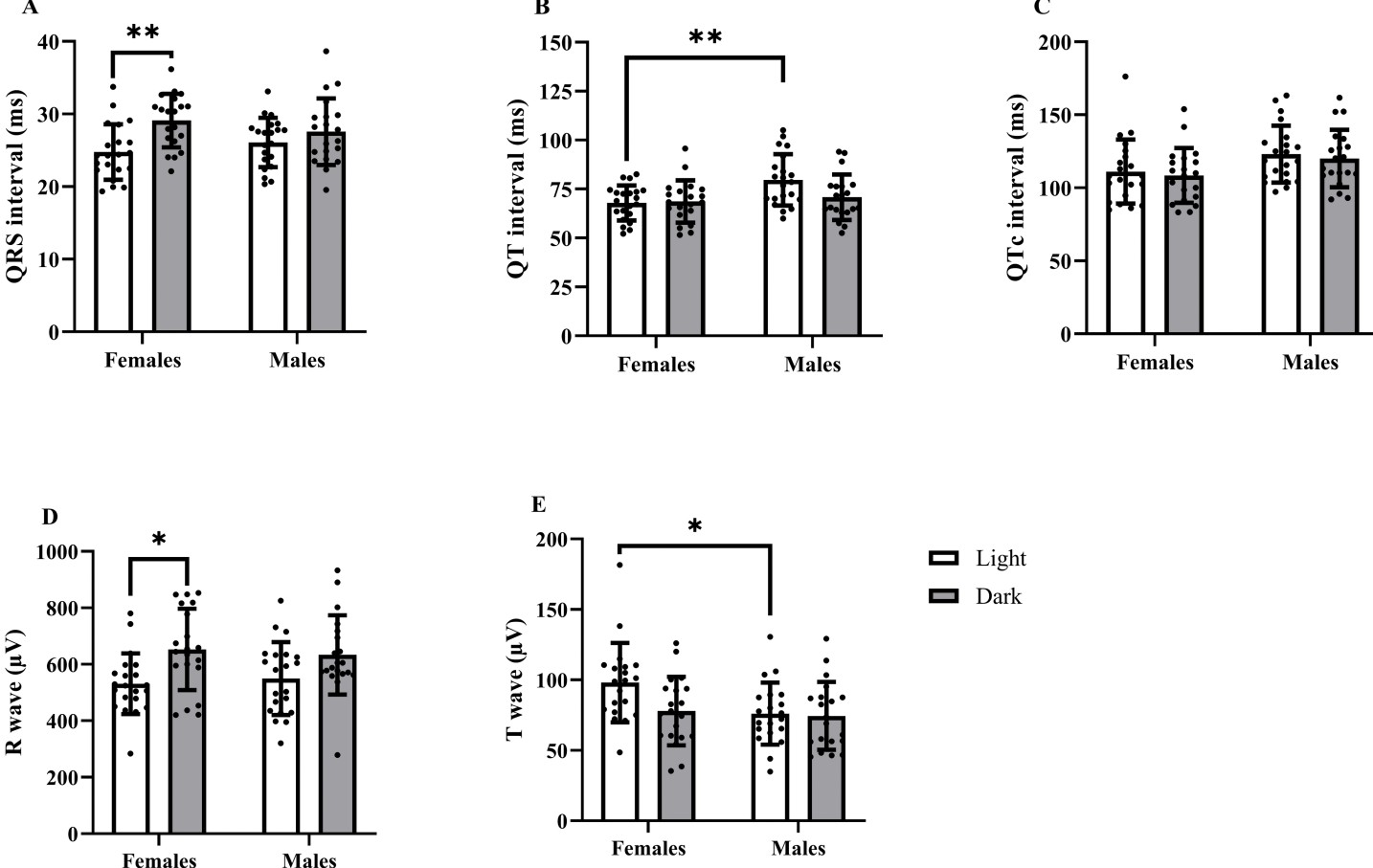

**Fig 4. Effect of tiletamine-zolazepam on the ventricular complex in Wistar rats depending on sex and LD cycle.** Differences in **(A)** QRS interval, **(B)** QT interval, **(C)** QTc interval, **(D)** R wave amplitude, and **(E)** T wave amplitude are presented as individual data points, with columns representing the mean ± SD for each group (white column – light period of the LD cycle; grey column – dark period of the LD cycle). Asterisks indicate the level of statistical significance according to Sidak's post hoc test: *$p<0.05$; **$p<0.01$; ***$p<0.001$; ****$p<0.0001$.

interval, which reflects both ventricular depolarisation and repolarisation, is reported to range from 69 to 71 ms in telemetry recordings in rats [30] and fluctuates between 57 and 76 ms under anaesthesia, depending on the type of anaesthetic used [1]. Abnormal prolongation of the QT interval may indicate disturbances in cardiac electrical activity caused, among other factors, by toxic exposure to external agents [1], with susceptibility to drug-induced arrhythmias differing between sexes [31]. In our experiments, males exhibited a prolonged QT interval during the light period of the LD cycle compared with females. The QTc interval, which corrects for HR changes, is considered a more objective indicator of ventricular depolarisation and repolarisation [1]. However, under tiletamine zolazepam, we observed only minor, statistically non-significant differences in the Bazett's formula-corrected QTc interval between the groups. Although alternative QTc correction formulas exist, the literature on rats remains inconsistent regarding the most appropriate method. Extensive use of different correction formulas may introduce significant errors [32]. As conducting a comprehensive comparison of different QTc corrections was beyond the scope of our study, we focused solely on Bazett's correction to ensure consistency and comparability. Nevertheless, further studies investigating the most suitable correction formulas for rats under anesthesia would be valuable. The T wave amplitude, which averages 150 µV in unanaesthetised Wistar rats [29] and 163.8 ± 58.4 µV

following ketamine-xylazine anaesthesia using wavelet transformation (delineation) of the ECG waveform [33], was lower under tiletamine-zolazepam and exhibited sex differences. Tiletamine-zolazepam likely affects ion channels or regulatory mechanisms involved in ventricular repolarisation, although further research is needed.

Although our results highlight the importance of sex and the LD cycle in cardiovascular studies using tiletamine-zolazepam, they should be interpreted with caution, taking into account individual variability, particularly among females. An important question is to what extent our data may have been influenced by the oestrous cycle and hormonal profile because different phases of the cycle can affect the autonomic regulation of the heart. The findings reported by Kuo et al. (2010) suggest that endogenous oestrogens play a crucial role in cardiovascular responses and vagal tone variability during the oestrous cycle [34]. Our previous study using tiletamine-zolazepam also indicated sex differences in HR variability in Wistar rats. In females, sympathetic and baroreflex activity predominated in both periods of the LD cycle, whereas parasympathetic activity predominated in males [24]. However, it is important to note that there is still a lack of high-quality studies involving direct measurement of ECG parameters in females under general anaesthesia across different phases of the oestrous cycle, as most experiments are conducted exclusively on males. Similarly, the impact of seasonal rhythms on ECG parameters in anaesthetised rats remains unclear, although their cardiac output shows seasonal variation [35], as does HR variability in small rodents [36]. In addition to sex and seasonal variation, factors such as breeding strain [1], age [37], body weight [38], stress [39], type and depth of anaesthesia, or body temperature [40] can also influence experimental outcomes. Future research should focus on addressing these limitations. The knowledge gained may contribute to a better understanding of the safety profile of anaesthetics, guide the optimal timing of experimental interventions to reduce cardiovascular risks, and improve the management of physiological responses during surgical procedures.

## Conclusion

Sex and LD cycle are key factors to consider when interpreting ECG parameters in experimental studies performed under tiletamine-zolazepam. HR in male Wistar rats showed a dependence on the LD cycle, whereas in females, differences between the light and dark periods were absent. The atrial complex (PR interval, P wave) was significantly modulated by sex, LD cycle, and their interaction—specifically in the case of the PR interval—while the amplitude of the P wave was unaffected by any of these factors. Similarly, the ventricular complex (QT, QTc, and QRS intervals, along with R and T waves) demonstrated dependence on sex and the LD cycle. However, the extent and nature of these effects varied across different ECG parameters. The duration of the QRS interval and the amplitude of the R wave changed depending on the LD cycle, whereas the QT and QTc intervals and T wave amplitude were more sensitive to sex. Nonetheless, post hoc analysis did not confirm any statistically significant differences between the experimental groups for the QTc interval.

## Supporting information

**S1 Table. Raw ECG data.** The file contains raw data extracted from LabChart 8 software. It includes individual values for all Wistar rats across the four experimental groups (males–light, females–light, males–dark, females–dark). Values marked as "NA" indicate parameters that could not be clearly identified from the recording and were therefore excluded from the corresponding analysis. ECG – electrocardiography.
(XLSX)

**S2 Fig. Representative ECG trace from a female Wistar rat anesthetized with tiletamine–zolazepam during the light period of the LD cycle.** The black waveform represents the averaged ECG trace, obtained by merging eight consecutive beats (individual beats are shown in green). Automatically detected ECG landmarks are indicated by vertical red/black lines: P start, P peak, P end, QRS start, QRS max, QRS end, T peak, and T end. Detection, analysis, and ECG trace export were performed using *LabChart 8* software. x-axis: time [s]; y-axis: ECG [μV].
(TIF)

**S3 Fig. Representative ECG trace from a female Wistar rat anesthetized with tiletamine–zolazepam during the dark period of the LD cycle.** The black waveform represents the averaged ECG trace, obtained by merging eight consecutive beats (individual beats are shown in green). Automatically detected ECG landmarks are indicated by vertical red/black lines: P start, P peak, P end, QRS start, QRS max, QRS end, T peak, and T end. Detection, analysis, and ECG trace export were performed using *LabChart 8* software. x-axis: time [s]; y-axis: ECG [μV].
(TIF)

**S4 Fig. Representative ECG trace from a male Wistar rat anesthetized with tiletamine–zolazepam during the light period of the LD cycle.** The black waveform represents the averaged ECG trace, obtained by merging eight consecutive beats (individual beats are shown in green). Automatically detected ECG landmarks are indicated by vertical red/black lines: P start, P peak, P end, QRS start, QRS max, QRS end, T peak, and T end. Detection, analysis, and ECG trace export were performed using *LabChart 8* software. x-axis: time [s]; y-axis: ECG [μV].
(TIF)

**S5 Fig. Representative ECG trace from a male Wistar rat anesthetized with tiletamine–zolazepam during the dark period of the LD cycle.** The black waveform represents the averaged ECG trace, obtained by merging eight consecutive beats (individual beats are shown in green). Automatically detected ECG landmarks are indicated by vertical red/black lines: P start, P peak, P end, QRS start, QRS max, QRS end, T peak, and T end. Detection, analysis, and ECG trace export were performed using *LabChart 8* software. x-axis: time [s]; y-axis: ECG [μV].
(TIF)

## Author contributions

**Conceptualization:** Soňa Grešová.

**Data curation:** Pavol Švorc.

**Formal analysis:** Viktória Novotná.

**Funding acquisition:** Viktória Novotná.

**Investigation:** Viktória Novotná, Soňa Grešová, Pavol Švorc.

**Methodology:** Pavol Švorc.

**Project administration:** Pavol Švorc.

**Resources:** Pavol Švorc.

**Software:** Viktória Novotná.

**Supervision:** Pavol Švorc.

**Validation:** Viktória Novotná, Soňa Grešová, Pavol Švorc.

**Visualization:** Viktória Novotná.

**Writing – original draft:** Viktória Novotná.

**Writing – review & editing:** Viktória Novotná, Soňa Grešová, Pavol Švorc.

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
