## [Decision Letter · Decision Letter 0]

16 Sep 2025

Dear Dr. Švorc,

We look forward to receiving your revised manuscript.

Kind regards,

Laxit K Bhatt

Academic Editor

PLOS ONE

Additional Editor Comments (if provided):

Reviewer #1:

Reviewer #2:

Reviewers' comments:

Reviewer's Responses to Questions

**Comments to the Author**

1. Is the manuscript technically sound, and do the data support the conclusions?

Reviewer #1: Partly

Reviewer #2: Yes

2. Has the statistical analysis been performed appropriately and rigorously?

Reviewer #1: I Don't Know

Reviewer #2: Yes

3. Have the authors made all data underlying the findings in their manuscript fully available?

Reviewer #1: Yes

Reviewer #2: Yes

4. Is the manuscript presented in an intelligible fashion and written in standard English?

Reviewer #1: No

Reviewer #2: Yes

Reviewer #1: Kapsdorferová et al. analyzed ECG parameters in Wostar rats dueing anesthesia with ZOLETIL (tiletamine and zolazepam for injection). Specifically, they studied the effects of gender and light/dark environment on the parameters. These data may be important. However, the puopose of this study is unclear and the study was not well designed.

#1 ZOLETIL is a brand name and it may not be suitable for the title.

#2 Though the authors claimed that rat model is close to human interms of cardiac functional studies. What is the evidence?

#3 LL68: The authors mentioned "The importance of this knowledge...." However, this mode is nothing to do with arrhythmia.

#4 The authors need to compare ZOLETIL with other anesthesia to measure ECG parameters. Did the author choose this med due to proarrhythmic?

#5 The QTc should be compared by other methods as well.

#6 It is difficult to see Table I and II. These should be combined. Data presentation is confusing in general.

#7 During AF, P waves should not be observed except AF/AFL. In addition, these models did not show AF.

#8 In human, it is well known that QT intervals are longer in female than male. The authors should show actual ECG traces. Did the authors check hormonal levels?

#9 I rellay do not understand why the authors can conclude that gender and L/D cycle are important in associations with arrhythmia. They did not show any consequences due to these difference in parameters.

Reviewer #2: Manuscript nicely drafted and thorough statistical analysis done vigorously and appropriately. Manuscript has been written in standard English. Author has made all data available in the manuscript for review.

**Do you want your identity to be public for this peer review?** For information about this choice, including consent withdrawal, please see our Privacy Policy

Reviewer #1: No

Reviewer #2: No

---

## [Author Response · Author response to Decision Letter 1]

2 Oct 2025

Response to Reviewers

We sincerely thank the academic editor and reviewers for their thoughtful comments and

constructive feedback on our manuscript titled „Chronobiological analysis of sex differences in

electrocardiographic parameters in spontaneously breathing Wistar rats under Zoletil anaesthesia“. We

have carefully considered all points raised and provide detailed responses below, indicating the

revisions made in the manuscript where applicable.

Response to Academic editor

COMMENT: -

RESPONSE: We appreciate the time and effort taken to review our work. We would also like to inform

the Academic Editor that the first author, Viktória Kapsdorferová, has changed her surname to Viktória

Novotná following her marriage. This change has been reflected in the revised manuscript.

Response to Reviewer #1

COMMENT: Kapsdorferová et al. analyzed ECG parameters in Wostar rats dueing anesthesia with

ZOLETIL (tiletamine and zolazepam for injection). Specifically, they studied the effects of gender and

light/dark environment on the parameters. These data may be important. However, the puopose of

this study is unclear and the study was not well designed.

RESPONSE: We appreciate the reviewer's critical comment. The rationale and objectives of our study

are clearly explained in the Introduction and Abstract. We examined whether ECG parameters differ

between the light and dark periods under Zoletil anaesthesia and between females and males under

Zoletil anaesthesia. This anaesthetic combination (tiletamine–zolazepam, Zoletil) is still rarely used in

rodent studies. Its cardiovascular effects are less well characterised compared to other anaesthetics,

even though rats are widely employed in cardiovascular research. Therefore, we believe that our data

may contribute to a better understanding of sex- and LD-dependent variability in rat cardiovascular

physiology under Zoletil anaesthesia and can be helpful for future experimental studies. It should be

emphasized that this study aimed to provide a numerical description of ECG parameter changes in

relation to sex and LD period. We did not aim to evaluate pathological phenomena, such as atrial

f

ibrillation, etc. This approach was considered sufficient to detect sex- and LD period-related

differences.

COMMENT: #1 ZOLETIL is a brand name and it may not be suitable for the title.

RESPONSE: We agree with the reviewer that generic names are generally preferred in scientific

manuscripts. Although the term Zoletil is commonly used in published studies (i.e., Cha et al., 2021;

Chengji et al., 2022; Debbarma et al., 2024; Lu et al., 2018; Morgan et al., 2012; Svorc et al., 2016), we

have carefully revised our manuscript and replaced the brand name with the generic formulation

t

iletamine–zolazepam where it was appropriate, including the title.

REFERENCES FOR THE RESPONSE:

Cha, K., Jeong, W. J., Kim, H. M., & So, B. H. (2021). Intravenous zoletil administration for the purpose

of suicide. Clinical and Experimental Emergency Medicine, 8(2), 149.

Chengji, W. A., Qing, C. H., Hui, G. O., Jue, W. A., Yinghan, W. A., Zhengye, G. U., Xu, B. A., & Ruling,

S. H. (2022). Anesthetic effects of different doses of Zoletil combined with Serazine hydrochloride on

C57BL/6J mice. Laboratory Animal and Comparative Medicine, 42(1), 31.

Debbarma, M., Bayan, H., Konwar, B., & Sarma, K. (2024). Exploring anaesthesia options in chickens:

Zoletil vs. Midazolam-Ketamine vs. Dexmedetomidine-Ketamine. The Indian Journal of Veterinary

Sciences and Biotechnology, 20(4), 119–123.

Lu, D. Z., Feng, X. J., Hu, K., Jiang, S., Li, L., Ma, X. W., & Fan, H. G. (2018). Inductive effect of Zoletil

on cystathionine β-synthase expression in the rat brain. International Journal of Biological

Macromolecules, 117, 1211–1215.

Morgan, D. R., Scobie, S., & Arthur, D. G. (2012). Evaluation of Zoletil and other injectable

anaesthetics for field sedation of brushtail possums (Trichosurus vulpecula). Animal Welfare, 21(4),

457–462.

Svorc, P., Bacova, I., Svorc Jr, P., Nováková, M., & Gresova, S. (2016). Zoletil anaesthesia in

chronobiological studies. Biological Rhythm Research, 47(1), 103–110.

COMMENT: #2 Though the authors claimed that rat model is close to human interms of cardiac

functional studies. What is the evidence?

RESPONSE: We would like to clarify that our manuscript does not state that the rat model is close to

human in terms of cardiac function. On page 3, line 50, we wrote: “Rats are considered the most

suitable model for studying the cardiovascular system.” This statement refers to their widespread use

in experimental cardiovascular research, not to direct translational applicability to humans. The term

“humans“ appears in our manuscript only in the following context: “P wave analysis has clinical

relevance in humans; however, high-quality reference data on its morphological and temporal changes

in rats are still lacking. Nevertheless, the absence of the P wave is associated with atrial fibrillation in

both humans and rats ” (page 14, lines 261–263). Our intention was not to generate results translatable

to humans, but rather to evaluate the differences in numerical ECG parameters between light and dark

periods and between male and female Wistar rats under Zoletil anaesthesia. We also acknowledge the

reviewer’s request for supporting evidence regarding the use of the rat model in cardiac functional

studies in relation to humans. As noted by Konopelski et al. (2016), despite certain interspecies

differences (e.g., the absence of a Q wave in most rat leads), there are fundamental similarities

between rat and human ECG. These similarities have supported the widespread use of rat ECG in basic

cardiovascular research, particularly for assessing cardiac function under physiological conditions and

in experimental models of cardiovascular disease. To clarify this potential misunderstanding, we have

added a short explanatory sentence with the above-mentioned reference to the Introduction section

of the revised manuscript.

REFERENCES FOR THE RESPONSE:

Konopelski P, Ufnal M. Electrocardiography in rats: a comparison to human. Physiological research.

2016 Sep 1;65(5):717.

COMMENT: #3 LL68: The authors mentioned "The importance of this knowledge...." However, this

mode is nothing to do with arrhythmia.

RESPONSE: We appreciate this valuable remark. As you pointed out, our publication did not address

arrhythmias, nor did we claim to do so. The purpose of the theoretical introduction was primarily to

emphasize the importance of understanding the effects of general anaesthesia on ECG parameters,

which subsequently serve as the basis for evaluating various pathologies, including arrhythmias (not in

our case, but in general). Although assessing pathological conditions was not the aim of our study, we

believe that our findings may be valuable for further experimental research on rats under Zoletil

anaesthesia that explicitly focuses on this issue.

COMMENT: #4 The authors need to compare ZOLETIL with other anesthesia to measure ECG

parameters. Did the author choose this med due to proarrhythmic?

RESPONSE: Our study was not designed to provide a comparative analysis of different anesthetics on

ECG parameters but rather to focus specifically on Zoletil. Since Zoletil is less frequently used in rat

research but more commonly applied in domestic and wild animals, its detailed effects on ECG

parameters in rats remain insufficiently characterized. Nonetheless, Zoletil may represent a suitable

anesthetic choice for future cardiovascular and chronobiological studies in rats due to its minimal

influence on vital functions and ECG rhythmicity. Accordingly, our study analyzed the effects of sex and

the light/dark cycle on ECG parameters in Wistar rats under Zoletil. We have slightly revised the

Introduction to clarify the rationale for choosing Zoletil. Regarding comparisons with other anesthetics,

our study focuses exclusively on Zoletil, as explained above. A broader comparative analysis of

anesthetics has already been addressed in our previous review, which examined ECG parameters under

various anesthetics across 123 studies (Svorc & SvorcJr., 2022).

REFERENCES FOR THE RESPONSE:

Svorc P, Svorc Jr P. General anesthesia and electrocardiographic parameters in in vivo experiments

involving rats. Physiological Research. 2022 Mar 11;71(2):177.

COMMENT: #5 The QTc should be compared by other methods as well.

RESPONSE: We appreciate the reviewer’s suggestion to compare QTc using multiple correction

formulas; however, the reviewer did not specify which alternative corrections should be included.

Moreover, the literature on rats is not consistent regarding which QTc correction is most appropriate

for this species. For example, Mulla et al. (2022) explicitly state: "In conclusion, conflicting data still

exist regarding the dependence between QT interval and heart rate in rodents. Multiple physiological

and technical complexities prevent clear conclusions based on currently available data. However, a

large body of evidence supports the notion that extensive use of different correction formulas may

introduce significant errors, and further systematic exploration of this issue would be of high value." In

our study, we applied Bazett’s correction, which is one of the most widely used in experimental and

clinical studies, including rat ECG research. Although other correction methods are available in

LabChart 8, applying them to the same ECG segments can yield highly variable results (see Response to Reviewers for Table and Fig). Such high variability reflects differences in the underlying formulas rather than actual

physiological changes and could introduce unnecessary bias. Therefore, we focused on Bazett’s

correction to ensure consistency and comparability with the existing literature. Conducting a complete

comparison of different QTc corrections was not the aim of our study; doing so would require the design

of an entirely new study. However, we have added a note in the Discussion acknowledging the existence

of alternative QTc correction formulas. We also highlighted that extensive use of different correction

formulas may introduce significant errors.

REFERENCES FOR THE RESPONSE:

Mulla W, Murninkas M, Levi O, Etzion Y. Incorrectly corrected? QT interval analysis in rats and mice.

Frontiers in Physiology. 2022 Oct 11;13:1002203.

COMMENT: #6 It is difficult to see Table I and II. These should be combined. Data presentation is

confusing in general.

RESPONSE: We appreciate the reviewer’s comment regarding the presentation of Tables I and II.

However, each table serves a distinct purpose: Table I summarizes the descriptive values only (mean ±

SD with 95% confidence intervals) for each parameter across all experimental groups, while Table II

presents the full statistical evaluation, including:

• Two-way ANOVA – p-values for the effects of sex, LD cycle period, and their interaction on all

ECG parameters in all groups.

• F-test (DFn, DFd) for between-group and within-group variability for sex, LD cycle period, and

their interaction.

• Sidak’s post hoc tests comparing all evaluated ECG parameters between experimental groups.

Given the complexity and volume of the data, combining the tables could reduce clarity rather than

enhance it. The rationale for using two-way ANOVA and Sidak’s post hoc tests is clearly explained in the

Methods section, allowing readers to understand the comparisons presented in Table II. For improved

readability, we have slightly revised the Results section by removing detailed numerical values from the

text, as these are fully reported in the tables, avoiding unnecessary repetition. Presenting the data in

this manner ensures that readers can comprehensively and transparently assess the effects of sex, LD

cycle, and their interaction on ECG parameters, as evaluated by two-way ANOVA, as well as any

differences between the experimental groups (females–light, males–light, females–dark, males–dark)

assessed using Sidak’s post hoc test.

COMMENT: #7 During AF, P waves should not be observed except AF/AFL. In addition, these models

did not show AF.

RESPONSE: We appreciate the reviewer’s comment. As correctly noted, our model did not show AF,

and this was not the purpose of our study. As mentioned above in our responses to Comments #1 and

#2, our study did not aim to evaluate pathological phenomena such as AF based on morphological

changes in the ECG waveform. Instead, we focused solely on the numerical description of ECG

parameter changes in relation to sex and the LD period. The text in the Discussion section referring to

the P wave only cites studies reporting these phenomena and was intended to provide readers with a

concise explanation of the physiological and pathophysiological meaning of the analyzed ECG

parameters. We have slightly revised this part of the Discussion to clarify what is derived from the

literature and what represents our own results.

COMMENT: #8 In humans, it is well known that QT intervals are longer in females than in males. The

authors should show actual ECG traces. Did the authors check hormonal levels?

RESPONSE: We thank the reviewer for this valuable comment. Indeed, in humans, females generally

have longer QT intervals than males due to reduced repolarization reserve, which increases their

susceptibility to polymorphic ventricular tachycardia of the torsades de pointes type, often associated

with drugs prolonging ventricular repolarization (Prajapati et al., 2022). In rats, however, most in vivo

experiments are performed exclusively in males, and therefore, valid comparative data on sex

differences are often lacking, including in cardiovascular and ECG-related studies, as emphasized also

in our previous work (Svorc & Svorc Jr., 2022). Regarding the present study with tiletamine–zolazepam,

we did not measure hormonal levels. As described in the Methods, results were averaged irrespective

of season and, in females, independently of the estrous cycle. We acknowledge this as a limitation of

our study in the Discussion section. Regarding actual ECG traces, we have now added representative

examples to the Supporting Information of the revised manuscript – one for each experimental group

(female–light, male–light, female–dark, male–dark). It should be noted that, due to the high

interindividual variability in rats even within the same experimental group, it was challenging to select

representative recordings that clearly display all significant effects. Nevertheless, we believe the

inclusion of these traces increases the transparency of our results.

REFERENCES FOR THE RESPONSE:

Prajapati C, Koivumäki J, Pekkanen-Mattila M, Aalto-Setälä K. Sex differences in heart: from basics

to clinics. European Journal of Medical Research. 2022 Nov 9;27(1):241.

Svorc P, Svorc Jr P. General anesthesia and electrocardiographic parameters in in vivo experiments

involving rats. Physiological Research. 2022 Mar 11;71(2):177.

COMMENT: #9 I really do not understand why the authors can conclude that gender and L/D cycle are

important in associations with arrhythmia. They did not show any consequences due to these

differences in parameters.

RESPONSE: We thank the reviewer for this comment. We are, however, unsure where in the manuscript

the reviewer found a statement suggesting that sex and the LD cycle directly affect arrhythmias in the

Conclusion section, as no specific line or page was indicated. The term “arrhythmia” appears four times

in the manuscript, in the following contexts:

1. Introduction: “Several studies also suggest an increased incidence of cardiovascular events in

the early morning hours, including stroke [11,12], myocardial infarction [13–15], ventricular

arrhythmias [16]…”

2. Experimental design: “For ECG recording, electrodes were applied subcutaneously, and the

depth of anaesthesia was reassessed using the same criteria—loss of motor activity (minimal

limb movement, changes in muscle tone) or cardiovascular responses to

---

## [Decision Letter · Decision Letter 1]

17 Nov 2025

Chronobiological analysis of sex differences in electrocardiographic parameters in spontaneously breathing Wistar rats under tiletamine-zolazepam anaesthesia  

PONE-D-25-38907R1

Dear Dr. Švorc,

We’re pleased to inform you that your manuscript has been judged scientifically suitable for publication and will be formally accepted for publication once it meets all outstanding technical requirements.

Kind regards,

Laxit K Bhatt

Academic Editor

PLOS ONE

Additional Editor Comments (optional):

Reviewers' comments:

Reviewer's Responses to Questions

**Comments to the Author**

Reviewer #1: All comments have been addressed

Reviewer #2: All comments have been addressed

2. Is the manuscript technically sound, and do the data support the conclusions?

Reviewer #1: Yes

Reviewer #2: Yes

3. Has the statistical analysis been performed appropriately and rigorously?

Reviewer #1: I Don't Know

Reviewer #2: Yes

4. Have the authors made all data underlying the findings in their manuscript fully available?

Reviewer #1: Yes

Reviewer #2: Yes

5. Is the manuscript presented in an intelligible fashion and written in standard English?

Reviewer #1: Yes

Reviewer #2: Yes

Reviewer #1: The authors successfully answered for my comments. Now that it is suitable to be published in PLoS ONE.

Reviewer #2: Manuscript may be accepted for publication. It may be a useful reference for preclinical research of pharmaceuticals.

**Do you want your identity to be public for this peer review?** For information about this choice, including consent withdrawal, please see our Privacy Policy

Reviewer #1: No

Reviewer #2: No

---

## [Editor Report · Acceptance letter]

PONE-D-25-38907R1

PLOS ONE

Dear Dr. Švorc,

I'm pleased to inform you that your manuscript has been deemed suitable for publication in PLOS ONE. Congratulations! Your manuscript is now being handed over to our production team.

Kind regards,

on behalf of

Mr. Laxit K Bhatt

Academic Editor

PLOS ONE